
# GPUTreeShap: massively parallel exact calculation of SHAP scores for tree ensembles

Rory Mitchell[1], Eibe Frank[2] and Geoffrey Holmes[2]

[1] Nvidia, Santa Clara, United States
[2] University of Waikato, Hamilton, New Zealand

## ABSTRACT

SHapley Additive exPlanation (SHAP) values (*Lundberg & Lee, 2017*) provide a game theoretic interpretation of the predictions of machine learning models based on Shapley values (*Shapley, 1953*). While exact calculation of SHAP values is computationally intractable in general, a recursive polynomial-time algorithm called TreeShap (*Lundberg et al., 2020*) is available for decision tree models. However, despite its polynomial time complexity, TreeShap can become a significant bottleneck in practical machine learning pipelines when applied to large decision tree ensembles. Unfortunately, the complicated TreeShap algorithm is difficult to map to hardware accelerators such as GPUs. In this work, we present GPUTreeShap, a reformulated TreeShap algorithm suitable for massively parallel computation on graphics processing units. Our approach first preprocesses each decision tree to isolate variable sized sub-problems from the original recursive algorithm, then solves a bin packing problem, and finally maps sub-problems to single-instruction, multiple-thread (SIMT) tasks for parallel execution with specialised hardware instructions. With a single NVIDIA Tesla V100-32 GPU, we achieve speedups of up to 19× for SHAP values, and speedups of up to 340× for SHAP interaction values, over a state-of-the-art multi-core CPU implementation executed on two 20-core Xeon E5-2698 v4 2.2 GHz CPUs. We also experiment with multi-GPU computing using eight V100 GPUs, demonstrating throughput of 1.2 M rows per second— equivalent CPU-based performance is estimated to require 6850 CPU cores.

# INTRODUCTION

Explainability and accountability are important for practical applications of machine learning, but the interpretation of complex models with state-of-the-art accuracy such as neural networks or decision tree ensembles obtained using gradient boosting is challenging. Recent literature (*Ribeiro, Singh & Guestrin, 2016*; *Selvaraju et al., 2017*; *Guidotti et al., 2018*) describes methods for "local interpretability" of these models, enabling the attribution of predictions for individual examples to component features. One such method calculates so-called SHapley Additive exPlanation (SHAP) values quantifying the contribution of each feature to a prediction. In contrast to other methods, SHAP values exhibit several unique properties that appear to agree with human intuition

Corresponding author
Rory Mitchell,
ramitchellnz@gmail.com

(*Lundberg et al., 2020*). Although exact calculation of SHAP values generally takes exponential time, the special structure of decision trees admits a polynomial-time algorithm. This algorithm, implemented alongside state-of-the-art gradient boosting libraries such as XGBoost (*Chen & Guestrin, 2016*) and LightGBM (*Ke et al., 2017*), enables complex decision tree ensembles with state-of-the-art performance to also output interpretable predictions.

However, despite improvements to algorithmic complexity and software implementation, computing SHAP values from tree ensembles remains a computational concern for practitioners, particularly as model size or size of test data increases: generating SHAP values can be more time-consuming than training the model itself. We address this problem by reformulating the recursive TreeShap algorithm, taking advantage of parallelism and increased computational throughput available on modern GPUs. We provide an open source module named GPUTreeShap implementing a high throughput variant of this algorithm using NVIDIA's CUDA platform. GPUTreeShap is integrated as a backend to the XGBoost library, providing significant improvements to runtime over its existing multicore CPU-based implementation.

## BACKGROUND

In this section, we briefly review the definition of SHAP values for individual features and the TreeShap algorithm for computing these values from decision tree models. We also review an extension of SHAP values to second-order interaction effects between features.

### SHAP values

SHAP values are defined as the coefficients of the following additive surrogate explanation model $g$, a linear function of binary variables

$$g(z') = \phi_0 + \sum_{i=1}^{M} \phi_i z_i' \tag{1}$$

where $M$ is the number of features, $z' \in \{0, 1\}^M$, and $\phi_i \in \mathbb{R}$. $z_{i'}$ indicates the presence of a given feature and $\phi_i$ its relative contribution to the model output. The surrogate model $g$ $(z')$ is a *local* explanation of a prediction $f(x)$ generated by the model for a feature vector $x$, meaning that a unique explanatory model may be generated for any given $x$. SHAP values are defined by the following expression:

$$\phi_i = \sum_{S \subseteq M \setminus \{i\}} \frac{|S|!(|M| - |S| - 1)!}{|M|!} [f_{S \cup \{i\}}(x) - f_S(x)] \tag{2}$$

where $M$ is the set of all features and $f_S(x)$ describes the model output restricted to feature subset $S$. Equation (2) considers all possible subsets, and so has runtime exponential in the number of features.

We consider models that are decision trees with binary splits. Given a trained decision tree model $f$ and data instance $x$, it is not necessarily clear how to restrict model output $f(x)$ to feature subset $S$—when feature $j$ is not present in subset $S$ along a given branch of the tree, and a split condition testing $j$ is encountered, then how do we choose which path

to follow to obtain a prediction for $x$? *Lundberg et al. (2020)* define a conditional expectation for the decision tree model $E[f(x)|x_S]$, where the split condition on feature $j$ is represented by a Bernoulli random variable with distribution estimated from the training set used to build the model. In effect, when a decision tree branch is encountered, and the feature to be tested is not in the active subset $S$, we take the output of both the left and right branch. More specifically, we use the proportion of weighted instances that flow down the left or right branch during model training as the estimated probabilities for the Bernoulli variable. This process is also how the C4.5 decision tree learner deals with missing values (*Quinlan, 1993*). It is referred to as "cover weighting" in what follows.

Given this interpretation of missing features, *Lundberg et al. (2020)* give a polynomial-time algorithm for efficiently solving Eq. (2), named TreeShap. The algorithm exploits the specific structure of decision trees: the model is additive in the contribution of each leaf. Equation (2) can thus be independently evaluated for each unique path from root to leaf node. These unique paths are then processed using a quadratic-time dynamic programming algorithm. The intuition of the algorithm is to keep track of the proportion of all feature subsets that flow down each branch of the tree, weighted according to the length of each subset $|S|$, as well as the proportion that flow down the left and right branches when the feature is missing.

We reproduce the recursive polynomial-time TreeSHAP algorithm as presented in *Lundberg et al. (2020)* in Algorithm 1, where $m$ is a list representing the path of unique features split on so far. Each list element has four attributes: $d$ is the feature index, $z$ is the fraction of paths that flow through the current branch when the feature is not present, $o$ is the corresponding fraction when the feature is present, and $w$ is the proportion of feature subsets of a given cardinality that are present. The decision tree is represented by the set of lists $\{v, a, b, t, r, d\}$, where each list element corresponds to a given tree node, with $v$ containing leaf values, $a$ pointers to the left children, $b$ pointers to the right children, $t$ the split condition, $r$ the weights of training instances, and $d$ the feature indices. The FINDFIRST function returns the index of the first occurrence of a feature in the list $m$, or a null value if the feature does not occur.

At a high level, the algorithm proceeds by stepping through a path in the decision tree of depth $D$ from root to leaf. According to Eq. (2), we have a different weighting for the size of each feature subset, although we can accumulate feature subsets of the same size together. As the algorithm advances down the tree, it calls the method EXTEND, taking a new feature split and accumulating its effect on all possible feature subsets of length $1, 2, \ldots$ up to the current depth. The UNWIND method is used to undo addition of a feature that has been added to the path *via* EXTEND. UNWIND and EXTEND are commutative and can be called in any order. UNWIND may be used to remove duplicate feature occurrences from the path and to compute the final SHAP values. When the recursion reaches a leaf, the SHAP values $\phi_i$ for each feature present in the path are computed by calling UNWIND on feature $i$ (line 7), temporarily removing it from the path; then, the overall effect of switching that feature on or off is adjusted by adding the appropriate term to $\phi_i$.

**Algorithm 1** TreeShap.

1: **function** TS(x, tree)

2:     $\phi$ = array of $len(x)$ zeroes

3:     **function** RECURSE($j$, $m$, $p_z$, $p_o$, $p_i$)

4:         $m$ = EXTEND($m$, $p_z$, $p_o$, $p_i$)

5:         **if** $v_j$ == *leaf* **then**

6:             **for** $i \leftarrow 2$ to $len(m)$ **do**

7:                 $w = sum(\text{UNWIND}(m, i).w)$

8:                 $\phi_{m_i.d} = \phi_{m_i.d} + w(m_i.o - m_i.z)v_j$

9:         **else**

10:            $h, c = x_{d_j} \leq t_j?(a_j, b_j) : (b_j, a_j)$

11:            $i_z = i_o = 1$

12:            $k = \text{FINDFIRST}(m.d, d_j)$

13:            **if** $k \neq nothing$ **then**

14:                $i_z, i_o = (m_k.z, m_k.o)$

15:                $m = \text{UNWIND}(m, k)$

16:            RECURSE($h$, $m$, $i_z r_h / r_j$, $i_o$, $d_j$)

17:            RECURSE($c$, $m$, $i_z r_c / r_j$, $0$, $d_j$)

18:     **function** EXTEND($m$, $p_z$, $p_o$, $p_i$)

19:         $l = len(m)$

20:         $m = copy(m)$ {$m$ is copied so recursions down other branches are not affected.}

21:         $m_{l+1}.(d, z, o, w) = (p_i, p_z, p_o, l = 0 \, ? \, 1 : 0)$

22:         **for** $i \leftarrow l$ to 1 **do**

23:             $m_{i+1}.w = m_{i+1}.w + p_o \cdot m_i.w \cdot i/(l + 1)$

24:             $m_i.w = p_z \cdot m_i.w \cdot (l + 1 - i)/(l + 1)$

25:         **return** $m$

26:     **function** UNWIND($m$, $i$)

27:         $l = len(m)$

28:         $n = m_l.w$

29:         $m = copy(m_{1 \ldots l-1})$

30:         **for** $j \leftarrow l - 1$ to 1 **do**

31:             **if** $m_i.o \neq 0$ **then**

32:                 $t = m_j.w$

33:                 $m_j.w = n \cdot l/(j \cdot m_i.o)$

34:                 $n = t - m_j.w \cdot m_i.z \cdot (l - j)/l$

35:             **else**

36:                 $m_j.w = (m_j.w \cdot l)/(m_i.z \cdot (l - j))$

37:         **for** $j \leftarrow i$ to $l - 1$ **do**

| Algorithm 1 (continued) |
| --- |
| 38:        $m_j.(d, z, o) = m_{j+1}.(d, z, o)$ |
| 39:      **return** $m$ |
| 40:     RECURSE(1, [], 1, 1, 0) |
| 41:     **return** $\phi$ |

Given an ensemble of $T$ decision trees, Algorithm 1 has time complexity $O(TLD^2)$, using memory $O(D^2 + M)$, where $L$ is the maximum number of leaves for each tree, $D$ is the maximum tree depth, and $M$ the number of features (*Lundberg et al., 2020*). In this paper, we reformulate Algorithm 1 for massively parallel GPUs.

## SHAP interaction values

In addition to the first-order feature relevance metric defined above, *Lundberg et al. (2020)* also provide an extension of SHAP values to second-order relationships between features, termed SHAP Interaction Values. This method applies the game-theoretic SHAP interaction index (*Fujimoto, Kojadinovic & Marichal, 2006*), defining a matrix of interactions as

$$\phi_{i,j} = \sum_{S \subseteq M \setminus \{i,j\}} \frac{|S|!(M - |S| - 2)!}{2(M - 1)!} \nabla_{ij}(S) \tag{3}$$

for $i \neq j$, where

$$\nabla_{ij}(S) = f_{S \cup \{i,j\}}(x) - f_{S \cup \{i\}}(x) - f_{S \cup \{j\}}(x) + f_S(x) \tag{4}$$
$$= f_{S \cup \{i,j\}}(x) - f_{S \cup \{j\}}(x) - [f_{S \cup \{i\}}(x) - f_S(x)] \tag{5}$$

with diagonals

$$\phi_{i,i} = \phi_i - \sum_{j \neq i} \phi_{i,j}. \tag{6}$$

Interaction values can be efficiently computed by connecting Eqs. (5) to (2), for which we have the polynomial time TreeShap algorithm. To compute $\phi_{i,j}$, TreeShap should be evaluated twice for $\phi_i$, where feature $j$ is alternately considered fixed to present and not present in the model. To evaluate TreeShap for a unique path conditioning on $j$, the path is extended as normal, but if feature $j$ is encountered, it is *not included* in the path (the dynamic programming solution is not extended with this feature, instead skipping to the next feature). If $j$ is considered not present, the resulting $\phi_i$ is weighted according to the probability of taking the left or right branch (cover weighting) at a split on feature $j$. If $j$ is considered present, we evaluate the decision tree split condition $x_j < t_j$ and discard $\phi_i$ from the path not taken.

To compute interaction values for all pairs of features, TreeShap can be evaluated $M$ times, leading to time complexity of $O(TLD^2M)$. Interaction values are challenging to compute in practice, with runtimes and memory requirements significantly larger than decision tree induction itself. In "Computing SHAP Interaction Values", we show how to

reformulate this algorithm to the GPU and how to improve runtime to $O(TLD^3)$ (tree depth $D$ is normally much smaller than the number of features $M$ present in the data).

## GPU computing

GPUs are massively parallel processors optimised for throughput, in contrast to conventional CPUs, which optimise for latency. GPUs in use today consist of many processing units with single-instruction, multiple-thread (SIMT) lanes that very efficiently execute a group of threads operating in lockstep. In modern NVIDIA GPUs such as the ones we use for the experiments in this paper, these processing units, called "streaming multiprocessors" (SMs), have 32 SIMT lanes, and the corresponding group of 32 threads is called a "warp". Warps are generally executed on SMs without order guarantees, enabling latency in warp execution (*e.g.*, from global memory loads) to be hidden by switching to other warps that are ready for execution (*NVIDIA Corporation, 2020*).[1]

Large speed-ups in the domain of GPU computing commonly occur when the problem can be expressed as a balanced set of vector operations with minimal control flow. Notable examples are matrix multiplication (*Fatahalian, Sugerman & Hanrahan, 2004*; *Hall, Carr & Hart, 2003*; *Jiang & Snir, 2005*), image processing (*Fang et al., 2005*; *Moreland & Angel, 2003*), deep neural networks (*Perry, Prosper & Meyer-Baese, 2014*; *Coates et al., 2013*; *Chetlur et al., 2014*), and sorting (*Green, McColl & Bader, 2012*; *Satish et al., 2010*). Prior work exists on decision tree induction (*Sharp, 2008*; *Mitchell & Frank, 2017*; *Zhang, Si & Hsieh, 2017*; *Dorogush, Ershov & Gulin, 2018*) and inference (*Sharp, 2012*) on GPUs, but we know of no prior work on tree interpretability specifically tailored to GPUs. Related work also exists on solving dynamic programming type problems (*Liu et al., 2006*; *Steffen, Giegerich & Giraud, 2010*; *Boyer, El Baz & Elkihel, 2012*), but dynamic programming is a broad term, and the referenced works discuss significantly different problem sizes and applications (*e.g.*, Smith-Waterman for sequence alignment).

In "GPUTreeShap", we discuss a unique approach to exploiting GPU parallelism, different from the above-mentioned works due to the unique characteristics of the TreeShap algorithm. In particular, our approach efficiently deals with large amounts of branching and load imbalance that normally inhibits performance on GPUs, leading to substantial improvements over a state-of-the-art multicore CPU implementation.

## GPUTREESHAP

Algorithm 1 has properties that make it unsuitable for direct implementation on GPUs in a performant way. Conventional multi-threaded CPU implementations of Algorithm 1 achieve parallel work distribution by instances (*Chen & Guestrin, 2016*; *Ke et al., 2017*). For example, interpretability results for input matrix $X$ are computed by launching one parallel CPU thread per row (*i.e.*, data instance being evaluated). While this approach is embarrassingly parallel, CPU threads are different from GPU threads. If GPU threads in a warp take divergent branches, performance is reduced, as all threads must execute identical instructions when they are active (*Harris & Buck, 2005*). Moreover, GPUs can suffer from per-thread load balancing problems—if work is unevenly distributed between threads in a warp, finished threads stall until all threads in the warp are finished.

[1] AMD GPUs have similar basic processing units, called "compute units"; the corresponding term for a warp is "wavefront".

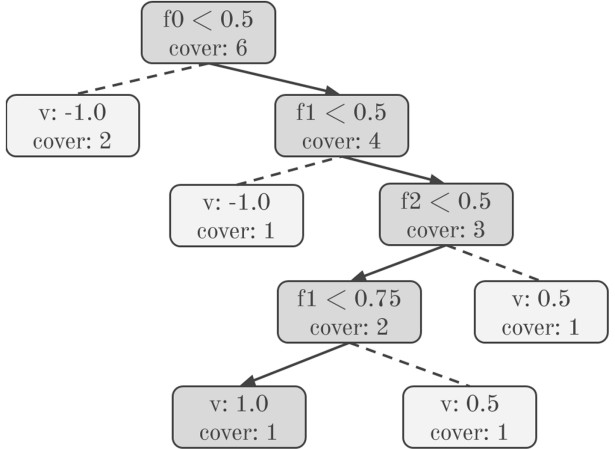

**Figure 1 Unique decision tree path.** Solid arrows indicate the path taken for an example test instance. Dashed lines indicate paths not taken.

Additionally, GPU threads are more resource-constrained than their CPU counterparts, having a smaller number of available registers due to limited per-SM resources. Excessive register usage results in reduced SM occupancy by limiting the number of concurrent warps. It also results in register spills to global memory, causing memory loads at significantly higher latency.

To mitigate these issues, we segment the TreeShap algorithm to obtain fine-grained parallelism, observing that each unique path from root to leaf in Algorithm 1 can be constructed independently because the $\phi_i$ obtained at each leaf are additive and depend only on features encountered on that unique path from root to leaf. Instead of allocating one thread per tree, we allocate a group of threads to cooperatively compute SHAP values for a given unique path through the tree. We launch one such group of threads for each (unique path, evaluation instance) pair, computing all SHAP values for this pair in a single GPU kernel. This method requires preprocessing to arrange the tree ensemble into a suitable form, avoid some less GPU-friendly operations of the original algorithm, and partition work efficiently across GPU threads. Our GPUTreeShap algorithm can be summarised by the following high-level steps:

1. Preprocess the ensemble to extract all unique decision tree paths.
2. Combine duplicate features along each path.
3. Partition path subproblems to GPU warps by solving a bin packing problem.
4. Launch a GPU kernel solving the dynamic programming subproblems in batch.

These steps are described in more detail below.

## Extract paths

Figure 1 shows a decision tree model, highlighting a unique path from root to leaf. The SHAP values computed by Algorithm 1 are simply the sum of the SHAP values from every unique path in the tree. Note that the decision tree model holds information about the

**Listing 1 Path element structure.**

```
struct PathElement {
    // Unique path index
    size_t path_idx;
    // Feature of this split, -1 is root
    int64_t feature_idx;
    // Range of feature value
    float feature_lower_bound;
    float feature_upper_bound;
    // Probability of following this path
    // when feature_idx is missing
    float zero_fraction;
    // Leaf weight at the end of path
    float v;
};
```

weight of training instances that flow down paths in the *cover* variable. To apply GPU computing, we first preprocess the decision tree ensemble into lists of path elements representing all possible unique paths in the ensemble. Path elements are represented as per Listing 1.

As paths share information that is represented in a redundant manner in the collection of lists representing a tree, reformulating trees increases memory consumption: assuming balanced trees, it increases space complexity from $O(TL)$ to $O(TDL)$ in the worst case. However, this additional memory consumption is not significant in practice.

Considering each path element, we use a lower and an upper bound to represent the range of feature values that can flow through a particular branch of the tree when the corresponding feature is present. For example, the root node in Fig. 1 has split condition $f_0 < 0.5$. Therefore, if the feature is present, the left branch from this node contains instances where $-\infty \le f_0 < 0.5$, and the right branch contains instances where $0.5 \le f_0 < \infty$. This representation is useful for the next preprocessing step, where we combine duplicate feature occurrences along a decision tree path.

Figure 2 shows two unique paths extracted from Fig. 1. An entire tree ensemble can be represented in this form. Crucially, this representation contains sufficient information to compute the ensemble's SHAP values.

## Remove duplicate features

Part of the complexity of Algorithm 1 comes from a need to detect and handle multiple occurrences of a feature in a single unique path. In Lines 12 to 15, the candidate feature of the current recursion step is checked against existing features in the path. If a previous occurrence is detected, it is removed from the path using the UNWIND function. The $p_z$

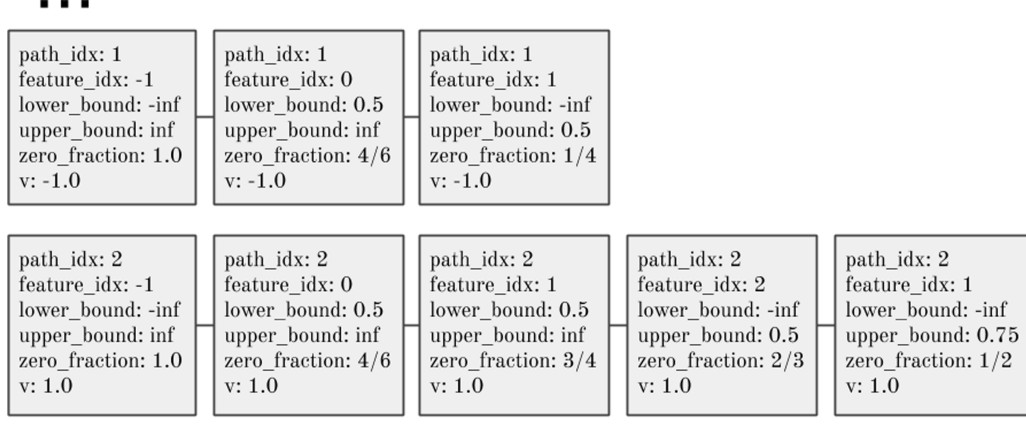

**Figure 2** **Two unique paths from the decision tree in Fig. 1.** The second path listed here corresponds to the highlighted path in Fig. 1, encoding bounds on the feature values for an instance that reaches this leaf, the leaf prediction value, and the conditional probability ('zero_fraction') of an instance meeting the split condition if the feature is unknown.

and $p_o$ values for the old and new occurrences of the feature are multiplied, and the path extended with these new values.

Unwinding previous features to deal with multiple feature occurrences in this manner is problematic for GPU implementation because it requires threads to cooperatively evaluate FINDFIRST and then UNWIND, introducing branching as well as extra computation. Instead, we take advantage of our representation of a tree ensemble in path element form, combining duplicate features into a single occurrence. To do this, recognise that a path through a decision tree from root to leaf represents a single hyperrectangle in the $M$ dimensional feature space, with boundaries defined according to split conditions along the path. The boundaries of the hyperrectangle may alternatively be represented with a lower and upper bound on each feature. Therefore, any number of decision tree splits over a single feature can be reduced to a single range, represented by these bounds. Moreover, note that the ordering of features within a path is irrelevant to the final SHAP values. As noted in *Lundberg et al. (2020)*, the EXTEND and UNWIND functions defined in Algorithm 1 are commutative; therefore, features may be added to or removed from a path in any order, and we can sort unique path representations by feature index, combining consecutive occurrences of the same feature into a single path element.

**Bin packing for work allocation**

Each unique path sub-problem identified above is mapped to GPU warps for hardware execution. A decision tree ensemble contains $L$ unique paths, where $L$ is the number of leaves, and each path has length between 1 and maximum tree depth $D$. To maximise throughput on the GPU, it is important to maximise utilisation of the processing units by saturating them with threads to execute. In particular, given a 32-thread warp, multiple paths may be resident and executed concurrently on a single warp. It is also important to assign all threads processing the same decision tree path to the same warp as we wish to use

**Table 1 Bin packing time complexities and worst-case approximation ratios.**

| Algorithm | Time | $R_A$ |
|---|---|---|
| NF | $O(n)$ | 2.0 |
| FFD | $O(n\log n)$ | 1.222 |
| BFD | $O(n\log n)$ | 1.222 |

fast warp hardware intrinsics for communication between these threads and avoid synchronisation cost. Consequently, in our GPU algorithm, sub-problems are constrained to not overlap across warps. This implies that the maximum depth of a decision tree processed by our algorithm must be less than or equal to the GPU warp size of 32. Given that the number of nodes in a balanced decision tree increases exponentially with depth, and real-world experience showing $D \le 16$ in high-performance boosted decision tree ensembles almost always, we believe this to be a reasonable constraint.

To achieve the highest device utilisation, path sub-problems should be mapped to warps such that the total number of warps is minimised. Given the above constraint, this requires solving a bin packing problem. Given a finite set of items $I$, with sizes $s(i) \in Z^+$, for each $i \in I$, and maximum bin capacity $B$, $I$ must be partitioned into the disjoint sets $I_0, I_1, \ldots, I_K$ such that the sum of sizes in each set is less than $B$. The partitioning minimising $K$ is the optimal bin packing solution. In our case, the bin capacity, $B = 32$, is the number of threads per warp, and our item sizes, $s(i)$, are given by the unique path lengths from the tree ensemble. In general, finding the optimal packing is strongly NP-complete (*Garey & Johnson, 1979*), although there are heuristics that can achieve close to optimal performance in many cases. In "Evaluating Bin Packing Performance", we evaluate three standard heuristics for the off-line bin packing problem, *Next-Fit* (NF), *First-Fit-Decreasing* (FF), and *Best-Fit-Decreasing* (BFD), as well as a baseline where each item is placed in its own bin. We briefly describe these algorithms and refer the reader to *Martello & Toth (1990)* or *Coffman, Garey & Johnson (1997)* for a more in-depth survey.

*Next-Fit* is a simple algorithm, where only one bin is open at a time. Items are added to the bin as they arrive. If bin capacity is exceeded, the bin is closed and a new bin is opened. In contrast, *First-Fit-Decreasing* sorts the list of items by non-increasing size. Then, beginning with the largest item, it searches for the first bin with sufficient capacity and adds it to the bin. Similarly, *Best-Fit-Decreasing* also sorts items by non-increasing size, but assigns items to the feasible bin with the smallest residual capacity. FFD and BFD may be implemented in $O(n \log n)$ time using a tree data structure allowing bin updates and insertions in $O(\log n)$ operations (see *Johnson (1974)* for specifics).

Existing literature gives worst-case approximation ratios for the above heuristics. For a given set of items $I$, let $A(I)$ denote the number of bins used by algorithm $A$, and $OPT(I)$ be the number of bins for the optimal solution. The approximation ratio $R_A \le \frac{A(I)}{OPT(I)}$ describes the worst-case performance ratio for any possible $I$. Time complexities and approximation ratios for each of the three above bin packing heuristic are listed in Table 1, as per *Coffman, Garey & Johnson (1997)*.

As this paper concerns the implementation of GPU algorithms, we would ideally formulate the above heuristics in parallel. Unfortunately, the bin packing problem is known to be hard to parallelise. In particular, FFD and BFD are P-complete, indicating that it is unlikely that these algorithms may be sped up significantly *via* parallelism (*Anderson & Mayr, 1984*). An efficient parallel algorithm with the same approximation ratio as FFD/BFD is given in *Anderson, Mayr & Warmuth (1989)*, but the adaptation of this algorithm to GPU architectures is nontrivial and beyond the scope of this paper. Fortunately, as shown by our evaluation in "Bin Packing For Work Allocation", CPU-based implementations of the bin packing heuristics give acceptable performance for our task, and the main burden of computation still falls on the GPU kernels computing SHAP values in the subsequent step. We perform experiments comparing the three bin packing heuristics in terms of runtime and impact on efficiency for GPU kernels in "Evaluating Bin Packing Performance".

## The GPU kernel for computing SHAP values

Given a unique decision tree path extracted from a decision tree in an ensemble predictor, with duplicates removed, we allocate one path element per GPU thread and cooperatively evaluate SHAP values for each row in a test dataset $X$. The dataset $X$ is assumed to be queryable from the device. Listing 2 provides a simplified overview of the GPU kernel that is the basis of GPUTreeShap, further details can be found at https://github.com/rapidsai/gputreeshap. A single kernel is launched, parallelising computation of Shapley values across GPU threads in three dimensions:

1. Dataset rows.
2. Unique paths in tree model from root to leaf.
3. Elements in each unique path.

GPU threads are launched according to the solution of the bin-packing problem described in "Bin Packing For Work Allocation", which allocates threads efficiently to this unevenly sized, three-dimensional problem space. A contiguous thread group of size $\leq 32$ is launched and assigned to each dataset row and model path sub-problem.

To enable non-recursive GPU-based implementation of Algorithm 1, it remains to describe how to compute permutation weights for each possible feature subset with the EXTEND function (Line 4), as well as how to UNWIND each feature in the path and calculate the sum of permutation weights (Line 7). The EXTEND function represents a single step in a dynamic programming problem. In the GPU version of the algorithm, it processes a single path in a decision tree, represented as a list of path elements. As discussed above, all threads processing the same path are assigned to the same warp to enable efficient processing. Data dependencies between threads occur when each thread processes a single path element. Figure 3 shows the data dependency of each call to EXTEND on previous iterations when using GPU threads for the implementation. Each thread depends on its own previous result and the previous result of the thread to its "left".

**Listing 2 GPU kernel overview—threads are mapped to elements of a path sub-problem, then groups of threads are formed.** These small thread groups cooperatively solve dynamic programming problems, accumulating the final SHAP values using global atomics.

```
__device__ float GetOneFraction(
      const PathElement& e, DatasetT X, size_t row_idx) {
   // First element in path (bias term) is always zero
   if (e.feature_idx == −1) return 0.0;
   // Test the split
   // Does the training instance continue down this
   // path if the feature is present?
   float val = X.GetElement(row_idx, e.feature_idx);
   return val >= e.feature_lower_bound &&
      val < e.feature_upper_bound;
}
template <typename DatasetT>
__device__ float ComputePhi(
      const PathElement& e, size_t row_idx,
      const DatasetT& X,
      const ContiguousGroup& group,
      float zero_fraction) {
   float one_fraction = GetOneFraction(e, X, row_idx);
   GroupPath path(group, zero_fraction, one_fraction);
   size_t unique_path_length = group.size();
   // Extend the path
   for (auto unique_depth = 1ull;
        unique_depth < unique_path_length;
        unique_depth++) {
     path.Extend();
  }
   float sum = path.UnwoundPathSum();
   return sum * (one_fraction − zero_fraction) * e.v;
}
template <typename DatasetT, size_t kBlockSize,
        size_t kRowsPerWarp>
__global__ void ShapKernel(
   DatasetT X, size_t bins_per_row,
   const PathElement* path_elements,
   const size_t* bin_segments, size_t num_groups,
   float* phis) {
```

**Listing 2** (continued)

```
__shared__ PathElement s_elements[kBlockSize];
    PathElement& e = s_elements[threadIdx.x];
    // Allocate some portion of rows to this warp
    // Fetch the path element assigned to this
    // thread
    size_t start_row, end_row;
    bool thread_active;
    ConfigureThread<DatasetT, kBlockSize, kRowsPerWarp>(
        X, bins_per_row, path_elements,
        bin_segments, &start_row, &end_row, &e,
        &thread_active);
    if (!thread_active) return;
    float zero_fraction = e.zero_fraction;
    auto labelled_group =
        active_labeled_partition(e.path_idx);
    for (int64_t row_idx = start_row;
        row_idx < end_row; row_idx++) {
      float phi =
        ComputePhi(e, row_idx, X, labelled_group,
            zero_fraction);
    // Write results
    if (!e.IsRoot()) {
      atomicAdd(&phis[IndexPhi(
          row_idx, num_groups, e.group,
          X.NumCols(), e.feature_idx)],
        phi);
    }
  }
}
```

This dependency pattern leads to a natural implementation using warp shuffle instructions, where threads directly access registers of other threads in the warp at considerably lower cost than shared or global memory. Algorithm 2 shows pseudo-code for a single step of a parallel EXTEND function on the device. In pseudocode, we define a shuffle function analogous to the corresponding function in NVIDIA's CUDA language, where the first argument is the register to be communicated, and the second argument is the thread to fetch the register from—if this thread does not exist, the function returns 0, else it returns the register value at the specified thread index. In Algorithm 2,

---

**Algorithm 2  Parallel EXTEND.**

1:   **function** PARALLEL_EXTEND($m, p_z, p_o, p_i$)
2:       $l = len(m)$
3:       $m_{l+1}.(d, z, o, w) = (p_i, p_z, p_o, l = 0\ ?\ 1 : 0)$
4:       **for** $i \leftarrow 2$ to $l + 1$ **in parallel, do**
5:           $left\_w = \text{shuffle}(m_i.w, i - 1)$
6:           $m_i.w = m_i.w \cdot p_z \cdot (l + 1 - i)/(l + 1)$
7:           $m_i.w = m_i.w + p_o \cdot left\_w \cdot i/(l + 1)$
8:       **return** $m$

---

**Algorithm 3  Parallel UNWOUNDSUM.**

1:   **function** PARALLEL_UNWOUNDSUM($m, p_z, p_o, p_i$)
2:       $l = len(m)$
3:       $sum = []$ array of $l$ zeroes
4:       **for** $i \leftarrow 1$ to $l + 1$ **in parallel, do**
5:           $next = \text{shuffle}(m_i.w, l)$
6:           **for** $j \leftarrow l$ to $1$ **do**
7:               $w_j = \text{shuffle}(m_i.w, j)$
8:               $tmp = (next \cdot (l - 1) + 1)/j$
9:               $sum_i = sum_i + tmp \cdot p_o$
10:              $next = w_j - tmp \cdot (l - j) \cdot p_z/l$
11:              $sum_i = sum_i + (1 - p_o) \cdot w_j \cdot l/((l - j) \cdot p_z)$
12:      **return** $sum$

---

the shuffle function is used to fetch the element $m_i.w$ from the current thread's left neighbour if this neighbour exists, and otherwise returns 0.

Given the permutation weights for the entire path, it is also necessary to establish how to UNWIND the effect of each individual feature from the path to evaluate its relative contribution (Algorithm 1, Line 7). We distribute this task among threads, with each thread "unwinding" a unique feature. Pseudo-code for UNWOUNDSUM is given in Algorithm 3, where each thread $i$ is effectively undoing the EXTEND function for a given feature and returning the sum along the path. Shuffle instructions are used to fetch weights $w_j$ from other threads in the group. The result of UNWOUNDSUM is used to compute the final SHAP value as per Algorithm 1, Line 8.

### Computing SHAP interaction values

Computation of SHAP interaction values makes use of the same preprocessing steps as above, and the same basic kernel building blocks, except that the thread group associated with each row/path pair evaluates SHAP values multiple times, iterating over each unique

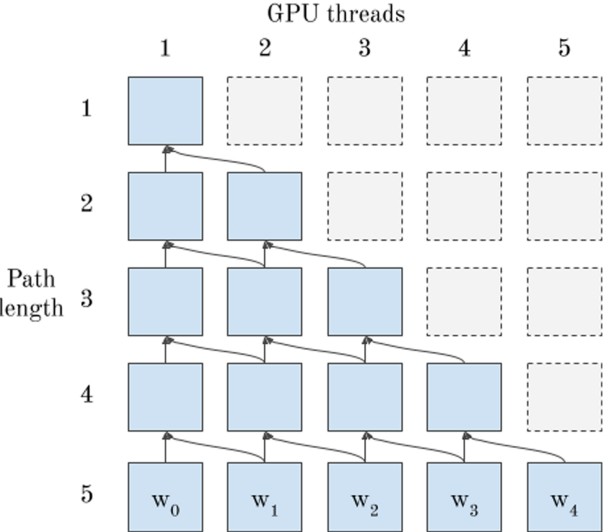

**Figure 3 Data dependencies of EXTEND—5 GPU threads communicate using warp shuffle intrinsics to solve a dynamic programming problem instance.**

feature and conditioning on that feature being fixed to present or not present respectively. There are some difficulties in conditioning on features with our algorithm formulation so far—conditioning on a feature $j$ requires ignoring it and not adding it to the active path. This introduces complexity when neighbouring threads are communicating *via* shuffle instructions (see Fig. 3). Each thread must adjust its indexing to skip over a path element being conditioned on. We found a more elegant solution is to swap a path element used for conditioning to the end of the path, then simply stop before adding it to the path (taking advantage of the fact that the ordering of path elements is irrelevant). Thus, to evaluate SHAP interaction values, we use a GPU kernel similar to the one used for computing per-feature SHAP values, except that we loop over each unique feature, conditioning on that feature as on or off.

One major difference that arises between our GPU algorithm and the CPU algorithm of *Lundberg et al. (2020)*, is that we can easily avoid conditioning on features that are not present in a given path. It is clear from Eq. (5) that $f_{S \cup \{i, j\}}(x) = f_{S \cup \{i\}}(x), f_{S \cup \{j\}}(x) = f_S(x)$ and $\nabla_{ij}(S) = 0$ if we condition on feature $j$ that is not present in the path. Therefore, our approach has runtime proportional to $O(TLD^3)$ instead of $O(TLD^2M)$ by exploiting the limited subset of possible feature interactions in a tree branch. This modification has a significant impact on runtime in practice (because, normally, $M \gg D$).

## EVALUATION

We train a range of decision tree ensembles using the XGBoost algorithm on the datasets listed in Table 2. Our goal is to evaluate a wide range of models representative of different real-world settings, from simple exploratory models to large ensembles of thousands of trees. For each dataset, we train a small, medium, and large variant by adjusting the number of boosting rounds (10, 100, 1,000) and maximum tree depth (3, 8, 16). The

**Table 2 Datasets used to train XGBoost models for Shapley value evaluation.** Rows refers to training rows, cols refers to number of features (excluding label).

| Name | Rows | Cols | Task | Classes | References |
|---|---|---|---|---|---|
| Covtype | 581,012 | 54 | Class | 8 | *Blackard (1998)* |
| Cal_housing | 20,640 | 8 | Regr | – | *Pace & Barry (1997)* |
| Fashion_mnist | 70,000 | 784 | Class | 10 | *Xiao, Rasul & Vollgraf (2017)* |
| Adult | 48,842 | 14 | Class | 2 | *Kohavi (1996)* |

**Table 3 XGBoost models used for evaluation.** Small, medium and large variants are created for each dataset.

| Model | Trees | Leaves | Max_depth |
|---|---|---|---|
| Covtype-small | 80 | 560 | 3 |
| Covtype-med | 800 | 113,888 | 8 |
| Covtype-large | 8,000 | 6,636,440 | 16 |
| Cal_housing-small | 10 | 80 | 3 |
| Cal_housing-med | 100 | 21,643 | 8 |
| Cal_housing-large | 1,000 | 3,317,209 | 16 |
| Fashion_mnist-small | 100 | 800 | 3 |
| Fashion_mnist-med | 1,000 | 144,154 | 8 |
| Fashion_mnist-large | 10,000 | 2,929,521 | 16 |
| Adult-small | 10 | 80 | 3 |
| Adult-med | 100 | 13,074 | 8 |
| Adult-large | 1,000 | 642,035 | 16 |

**Table 4 Details of Nvidia DGX-1 used for benchmarking.**

| Processor | Details |
|---|---|
| CPU | $2 \times$ 20-Core Xeon E5-2698 v4 2.2 GHz |
| GPU | $8 \times$ Tesla V100-32 |

learning rate is set to 0.01 to prevent XGBoost learning the model in the first few trees and producing only stumps in subsequent iterations. Using a low learning rate is also common in practice to minimise generalisation error. Other hyperparameters are left as default. Summary statistics for each model variant are listed in Table 3, and our testing hardware is listed in Table 4.

## Evaluating bin packing performance

We first evaluate the performance of the NF, FFD, and BFD bin packing algorithms from "Bin Packing For Work Allocation". We also include "none" as a baseline, where no packing occurs and each unique path is allocated to a single warp. All bin packing heuristics are single-threaded and run on the CPU. We report the execution time (in seconds), utilisation, and number of bins used ($K$). Utilisation is defined as $\frac{\sum_{i \in I} s(i)}{32K}$, the

total weight of all items divided by the bin space allocated, or for our purposes, the fraction of GPU threads that are active for a given bin packing. Poor bin packings waste space on each warp and underutilise the GPU.

Results are summarised in Table 5. "None" is clearly a poor choice, with utilisation between 0.1 and 0.3, with worse utilisation for smaller tree depths—for example, small models with maximum depth three allocate items of size three to warps of size 32. The simple NF algorithm often provides competitive results with fast runtimes, but it can lag behind FFD and BFD when item sizes are larger, exhibiting utilisation as low as 0.79 for *fashion_mnist-large*. FFD and BFD achieve better utilisation than NF in all cases, reflecting their superior approximation guarantees. Interestingly, FFD and BFD achieve the same efficiency on every example tested. We have verified that they can produce different packings on contrived examples, but there is no discernible difference for our application. FFD and BFD have longer runtimes than NF due to their $O(n \log n)$ time complexity. FFD is slightly faster than BFD because it uses a binary tree packed into an array, yielding greater cache efficiency, but its implementation is more complicated. In contrast, BFD is implemented easily using *std::set*.

Based on these results, we recommend BFD for its strong approximation guarantee, simple implementation, and acceptable runtime when packing jobs into batches for GPU execution. Its runtime is at most 1.6 s in our experiments, for our largest model (*covtype-large*) with 6.7 M items, and is constant with respect to the number of test rows because the bin packing occurs once per ensemble and is reused for each additional data point, allowing us to amortise its cost over improvements in end-to-end runtime from improved kernel efficiency. The gains in GPU thread utilisation from using BFD over NF directly translate into performance improvements, as fewer bins used means fewer GPU warps are launched. On our large size models, we see improvements in utilisation of 10.1%, 3.2%, 16.7% and 9.6% from BFD over NF. We use BFD in all subsequent experiments.

## Evaluating SHAP value throughput

We evaluate the performance of GPUTreeShap as a backend to the XGBoost library (*Chen & Guestrin, 2016*), comparing its execution time against the existing CPU implementation of Algorithm 1[2]. The baseline CPU algorithm is efficiently multithreaded using OpenMP, with a parallel for loop over all test instances. See https://github.com/dmlc/xgboost for exact implementation details for the baseline and https://github.com/rapidsai/gputreeshap for GPUTreeShap implementation details.

Table 6 reports the runtime of GPUTreeShap on a single V100 GPU compared to TreeShap on 40 CPU cores. Results are averaged over five runs and standard deviations are also shown. We observe speedups between 13 and 19× for medium and large models evaluated on 10,000 test rows. We observe little to no speedup for the small models as insufficient computation is performed to offset the latency of launching GPU kernels.

Figure 4 plots the time to evaluate varying numbers of test rows for the *cal_housing-med* model. We plot the average of five runs; the shaded area indicates the 95% confidence interval. This illustrates the throughput *vs.* latency trade-off for this particular model size.

[2] We do not benchmark against TreeShap implementations in the Python SHAP package or LightGBM because they are written by the same author, also in C++, and are functionally equivalent to XGBoost's implementation.

**Table 5 Bin packing performance.**

| Model | Alg | Time (s) | Utilisation | Bins |
|-------|-----|----------|-------------|------|
| Covtype-small | None | 0.0018 | 0.105246 | 560 |
| Covtype-small | NF | 0.0041 | 0.982292 | 60 |
| Covtype-small | FFD | 0.0064 | 0.998941 | 59 |
| Covtype-small | BFD | 0.0086 | 0.998941 | 59 |
| Covtype-med | None | 0.0450 | 0.211187 | 113,533 |
| Covtype-med | NF | 0.0007 | 0.913539 | 26,246 |
| Covtype-med | FFD | 0.0104 | 0.940338 | 25,498 |
| Covtype-med | BFD | 0.0212 | 0.940338 | 25,498 |
| Covtype-large | None | 0.0346 | 0.299913 | 6,702,132 |
| Covtype-large | NF | 0.0413 | 0.851639 | 2,360,223 |
| Covtype-large | FFD | 0.8105 | 0.952711 | 2,109,830 |
| Covtype-large | BFD | 1.6702 | 0.952711 | 2,109,830 |
| Cal_housing-small | None | 0.0015 | 0.085938 | 80 |
| Cal_housing-small | NF | 0.0025 | 0.982143 | 7 |
| Cal_housing-small | FFD | 0.0103 | 0.982143 | 7 |
| Cal_housing-small | BFD | 0.0001 | 0.982143 | 7 |
| Cal_housing-med | None | 0.0246 | 0.181457 | 21,641 |
| Cal_housing-med | NF | 0.0126 | 0.931429 | 4,216 |
| Cal_housing-med | FFD | 0.0016 | 0.941704 | 4,170 |
| Cal_housing-med | BFD | 0.0031 | 0.941704 | 4170 |
| Cal_housing-large | None | 0.0089 | 0.237979 | 3,370,373 |
| Cal_housing-large | NF | 0.0225 | 0.901060 | 890,148 |
| Cal_housing-large | FFD | 0.3534 | 0.933114 | 859,570 |
| Cal_housing-large | BFD | 0.8760 | 0.933114 | 859570 |
| Fashion_mnist-small | None | 0.0022 | 0.123906 | 800 |
| Fashion_mnist-small | NF | 0.0082 | 0.991250 | 100 |
| Fashion_mnist-small | FFD | 0.0116 | 0.991250 | 100 |
| Fashion_mnist-small | BFD | 0.0139 | 0.991250 | 100 |
| Fashion_mnist-med | None | 0.0439 | 0.264387 | 144,211 |
| Fashion_mnist-med | NF | 0.0008 | 0.867580 | 43,947 |
| Fashion_mnist-med | FFD | 0.0130 | 0.880279 | 43,313 |
| Fashion_mnist-med | BFD | 0.0219 | 0.880279 | 43,313 |
| Fashion_mnist-large | None | 0.0140 | 0.385001 | 2,929,303 |
| Fashion_mnist-large | NF | 0.0132 | 0.791948 | 1,424,063 |
| Fashion_mnist-large | FFD | 0.3633 | 0.958855 | 1,176,178 |
| Fashion_mnist-large | BFD | 0.8518 | 0.958855 | 1,176,178 |
| Adult-small | None | 0.0016 | 0.125000 | 80 |
| Adult-small | NF | 0.0023 | 1.000000 | 10 |
| Adult-small | FFD | 0.0061 | 1.000000 | 10 |
| Adult-small | BFD | 0.0060 | 1.000000 | 10 |
| Adult-med | None | 0.0050 | 0.229014 | 13,067 |

| Model | Alg | Time (s) | Utilisation | Bins |
|-------|-----|----------|-------------|------|
| Adult-med | NF | 0.0066 | 0.913192 | 3,277 |
| Adult-med | FFD | 0.0575 | 0.950010 | 3,150 |
| Adult-med | BFD | 0.1169 | 0.950010 | 3,150 |
| Adult-large | None | 0.0033 | 0.297131 | 642,883 |
| Adult-large | NF | 0.0035 | 0.858728 | 222,446 |
| Adult-large | FFD | 0.0684 | 0.954377 | 200,152 |
| Adult-large | BFD | 0.0954 | 0.954377 | 200,152 |

**Table 6 Speedups for V100 *vs.* 40 CPU cores on 10,000 test rows.**

| Model | CPU (s) | Std | GPU (s) | Std | Speedup |
|-------|---------|-----|---------|-----|---------|
| Covtype-small | 0.04 | 0.02 | 0.02 | 0.01 | 2.27 |
| Covtype-med | 8.25 | 0.07 | 0.45 | 0.03 | 18.23 |
| Covtype-large | 930.22 | 0.56 | 50.88 | 0.21 | 18.28 |
| Cal_housing-small | 0.01 | 0.01 | 0.01 | 0.01 | 0.96 |
| Cal_housing-med | 1.27 | 0.02 | 0.09 | 0.02 | 14.59 |
| Cal_housing-large | 315.21 | 0.30 | 16.91 | 0.34 | 18.64 |
| Fashion_mnist-small | 0.35 | 0.14 | 0.17 | 0.04 | 2.09 |
| Fashion_mnist-med | 15.10 | 0.07 | 1.13 | 0.08 | 13.36 |
| Fashion_mnist-large | 621.14 | 0.14 | 47.53 | 0.17 | 13.07 |
| Adult-small | 0.01 | 0.00 | 0.01 | 0.01 | 1.08 |
| Adult-med | 1.14 | 0.00 | 0.08 | 0.01 | 14.59 |
| Adult-large | 88.12 | 0.20 | 4.67 | 0.00 | 18.87 |

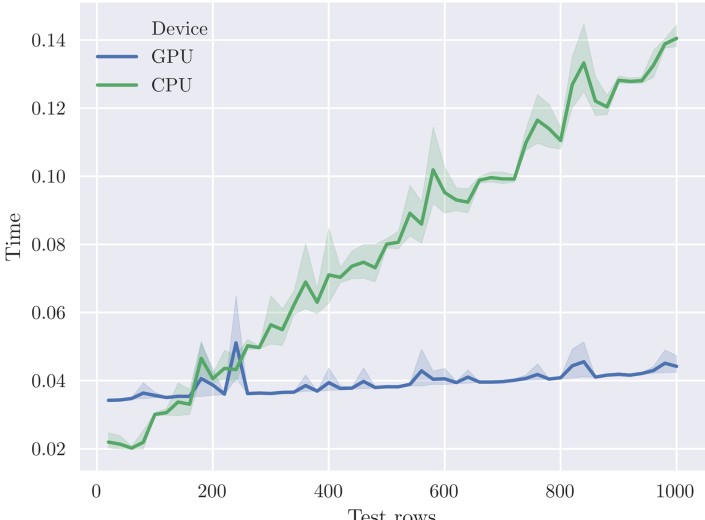

**Figure 4 The crossover point where the V100 GPU outperforms 40 CPU cores occurs at around 200 test rows for the *cal_housing-med* model.**

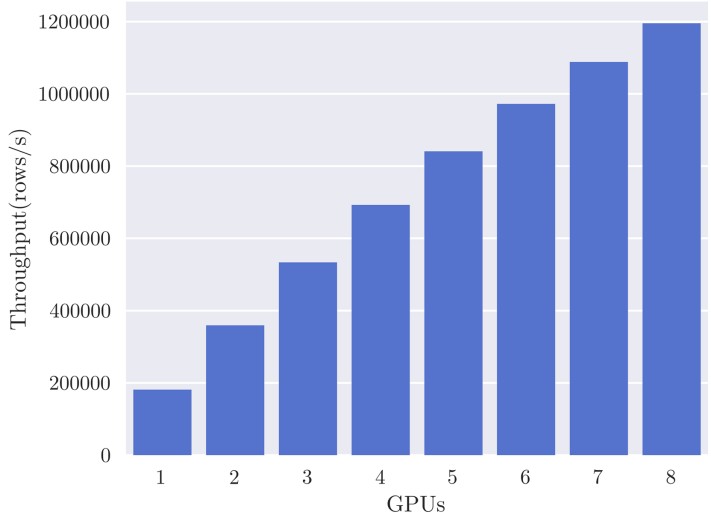

**Figure 5** **GPUTreeShap scales linearly with 8 V100 GPUs for the *cal_housing-med* model.**

The CPU is more effective for <180 test rows due to lower latency, but the throughput of the GPU is significantly higher at larger batch sizes.

SHAP value computation is embarrassingly parallel over dataset rows, so we expect to see linear scaling of performance with respect to the number of GPUs or CPUs, given sufficient data. We set the number of rows to 1 million and evaluate the effect of additional processors for the *cal_housing-med* model, measuring throughput in rows per second. Figure 5 reports throughput up to the eight GPUs available on the DGX-1 system, showing the expected close to linear scaling and reaching a maximum throughput of 1.2 M rows per second. Reported throughputs are from the average of five runs—error bars are too small to see due to relatively low variance. Figure 6 shows linear scaling with respect to CPU cores up to a maximum throughput of 7,000 rows per second. The shaded area indicates the 95% confidence interval from 5 runs. We speculate that the dip at 40 cores is due to contention with the operating system requiring threads for other system functions, and so ignore it for this scaling analysis. We can reasonably approximate from Fig. 6, using a throughput of 7,000 rows/s per 40 cores, that it would require 6850 Xeon E5-2698 v4 CPU cores, or 343 sockets, to achieve the same throughput as eight V100 GPUs for this particular model.

## SHAP interaction values

Table 7 compares single GPU *vs.* 40 core CPU runtime for SHAP interaction values. For this experiment, we lower the number of test rows to 200 due to the significantly increased computation time. Computing interaction values is challenging for datasets with larger numbers of features, in particular for *fashion_mnist* (785 features). Our GPU implementation achieves moderate speedups on *cal_housing* and *adult* due to the relatively low number of features; these speedups are roughly comparable to those obtained for standard SHAP values (Table 6). In contrast, for *covtype-large* and *fashion_mnist-large*, we

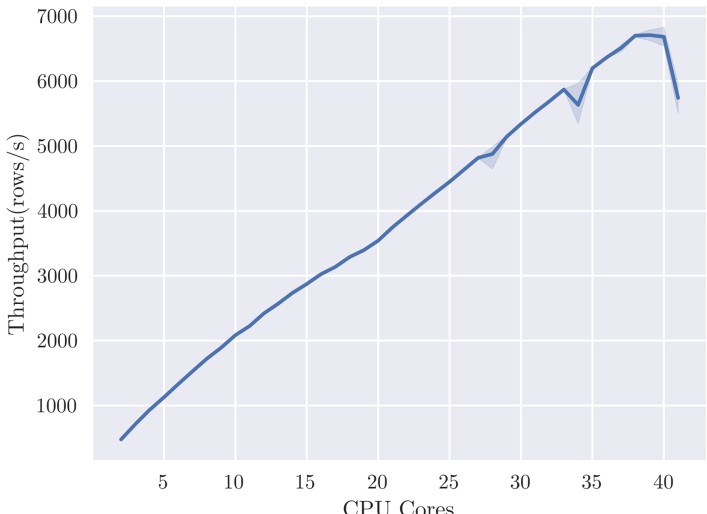

**Figure 6 TreeShap scales linearly with 40 CPU cores, but at significantly lower throughput than GPUTreeShap.**

**Table 7 Feature interactions—Speedups for V100 *vs.* 40 CPU cores on 200 test rows.**

| Model | CPU (s) | Std | GPU (s) | Std | Speedup |
|---|---|---|---|---|---|
| Covtype-small | 0.14 | 0.01 | 0.02 | 0.01 | 8.32 |
| Covtype-med | 21.50 | 0.32 | 0.19 | 0.02 | 114.41 |
| Covtype-large | 2,055.78 | 4.19 | 28.85 | 0.06 | 71.26 |
| Cal_housing-small | 0.01 | 0.00 | 0.01 | 0.00 | 1.44 |
| Cal_housing-med | 0.53 | 0.04 | 0.04 | 0.01 | 12.05 |
| Cal_housing-large | 93.67 | 0.28 | 8.55 | 0.04 | 10.96 |
| Fashion_mnist-small | 11.35 | 0.87 | 4.04 | 0.67 | 2.81 |
| Fashion_mnist-med | 578.90 | 1.23 | 4.91 | 0.71 | 117.97 |
| Fashion_mnist-large | 21,603.53 | 622.60 | 63.53 | 0.78 | 340.07 |
| Adult-small | 0.06 | 0.09 | 0.01 | 0.00 | 11.25 |
| Adult-med | 1.74 | 0.30 | 0.04 | 0.01 | 39.38 |
| Adult-large | 67.29 | 6.22 | 2.76 | 0.00 | 24.38 |

see speedups of 114× and 340×, in the most extreme case reducing runtime from 6 h to 1 min. This speedup comes from both the increased throughput of the GPU over the CPU and the improvements to algorithmic complexity due to omission of irrelevant features described in "Computing SHAP Interaction Values". Note that it may be possible to reformulate the CPU algorithm to take advantage of the improved complexity with similar preprocessing steps, but investigating this is beyond the scope of this paper.

## CONCLUSION

SHAP values have proven to be a useful tool for interpreting the predictions of decision tree ensembles. We have presented GPUTreeShap, an algorithm obtained by reformulating

the TreeShap algorithm to enable efficient computation on GPUs. We exploit warp-level parallelism by cooperatively evaluating dynamic programming problems for each path in a decision tree ensemble, thus providing massive parallelism for large ensemble predictors. We have shown how standard bin packing heuristics can be used to effectively schedule problems at the warp level, maximising GPU utilisation. Additionally, our rearrangement leads to improvement in the algorithmic complexity when computing SHAP interaction values, from $O(TLD^2M)$ to $O(TLD^3)$. Our library GPUTreeShap provides significant improvement to SHAP value computation over currently available software, allowing scaling onto one or more GPUs, and reducing runtime by one to two orders of magnitude.

### Funding
The authors received no funding for this work.

### Competing Interests
Eibe Frank is an Academic Editor for PeerJ. Rory Mitchell is employed by Nvidia Corporation.

### Author Contributions
- Rory Mitchell conceived and designed the experiments, performed the experiments, performed the computation work, prepared figures and/or tables, and approved the final draft.
- Eibe Frank analyzed the data, authored or reviewed drafts of the paper, and approved the final draft.
- Geoffrey Holmes analyzed the data, authored or reviewed drafts of the paper, and approved the final draft.

### Data Availability
The source code is available at GitHub: https://github.com/rapidsai/gputreeshap.
The public datasets are available at:
- Forest cover type: https://archive.ics.uci.edu/ml/datasets/covertype.
- California housing: https://scikit-learn.org/stable/modules/generated/sklearn.datasets.fetch_california_housing.html.
- Adult: https://archive.ics.uci.edu/ml/datasets/adult.
- Fashion-mnist: https://github.com/zalandoresearch/fashion-mnist

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
