# Peer review of "GPUTreeShap: massively parallel exact calculation of SHAP scores for tree ensembles"

_PeerJ Computer Science, doi:10.7717/peerj-cs.880_

## Round 0.1 · original submission · Minor Revisions

· Academic Editor

Minor Revisions

Please address the comments of the reviewers

·

Basic reporting

The submission is well-written and structured appropriately. The key ideas of the research are communicated clearly. The literature review and background are sufficient to understand the article. The results are presented clearly and are supported by the availability of source-code and datasets.

Experimental design

There is a clear research question in that there is no (to the best of my knowledge) previous efficient implementation for computing the TreeShap algorithm on GPU architectures. This knowledge gap is filled by presenting and evaluating an optimised and interesting GPU implementation.

The conclusions of the article are supported by the results. The experiments are well-designed and would be straightforward to replicate (very easily given the availability of the datasets and source-code used).

Validity of the findings

As above, the conclusions of the the article are supported by the results.

Additional comments

The GPU implementation described by this article presents both an insightful rearranging of the algorithm to make it suitable for GPU architectures as well as optimisation that shows a thorough understanding of the architectural features available.

·

Basic reporting

The writing is generally good. The introduction, literature review, and background are sufficient and well-written.

Some minor issues are:
1. In Figure 1, please explain in the caption or text what the left/right arrows, solid/dash arrows mean for more clarification.
2. In Figure 2, it is better to indicate how these two unique paths correspond to the two leaf nodes (and which two) in Figure 1.
3. Please double-check Lines 22-24 of Algorithm 1, as well as related parts (Algorithm 2) and your code implementation. They are inconsistent with either (Lundberg et al, 2020) or their earlier publication (https://arxiv.org/abs/1802.03888). If it is not an error, can you confirm or explain the difference?
4. In Section 3.2, Line 184, please double-check which of "Line 12" or "Line 13" (of Algorithm 1) is more suitable for the context.
5. In Table 2, "rows" and "cols" need explanation. Does a column mean a feature? Does it include the class label? Why does fashion_mnist contain 785 columns while the image size is 28x28=784? For cal_housing, does it include longitude and latitude features?
6. In Line 305, LaTeX symbol \gg may be considered for "much greater than".
7. In Line 319, are s(i) and I defined before?
8. For easier reading/comparison, is it possible to reorganize Table 5 so that models and algs are in different dimensions (e.g. models in rows, algs in columns)? If not possible due to limited space, can horizontal split lines be added to separate different models?
9. Please double-check the caption of Table 7, as it is said in Line 373 of the main text that the number of test rows was reduced to 200 but the caption said 10000.

Experimental design

The research and experiments are original and carried out well, supporting the original findings in general, but here is one issue/suggestion to enhance the experiments.

In Line 339, the authors claimed that greater utilization directly translates to performance improvement. Is it possible to add additional experiments that support this statement? The authors may consider comparing the runtime speedups of BFD vs NF, or BFD vs None.

Validity of the findings

Source code is publicly available. Refer to issue 3 in basic reporting. Otherwise no comment.

Additional comments

Can GPUTreeSHAP result in exact SHAP values and interaction values, or just provide their approximations due to approximated heuristics in bin packing? If they are approximations, what is the error?

·

Basic reporting

This paper is about a GPU implementation of the well-known machine learning interpretability method SHAP. The method is recursive polynomial time and until now was only possible to use it on relatively small datasets. The proposed GPU implementation achieves up to 19x speedups. I have not seen any similar papers.
The paper is well-written and includes all the required sections. The proposed GPU algorithm is described using enough details and pseudocode is included in the paper.
The code is available on github and the datasets are available online.
Things to improve:
- In section 2.3 GPU Computing, the authors could mention that both the LightGBM and Catboost have GPU implementations.
- In section 5 Conclusions, the authors could comment on extending the proposed algorithms to the other gradient boosting algorithms. Is the proposed method good for solving other problems?
- The abstract says 19x speedups, but section 4.2 says between 13-18x for medium and large models.
- Please add a description of the multi-core CPU implementation.
- The paper mentions the theoretical upper bounds of memory consumption, but no experimental memory consumption number are reported in the paper.

Experimental design

The authors evaluate the proposed method using several sets of parameters for the XGBoost trees. The method is evaluated in 4 different datasets. I wish the authors would have run experiment with even larger datasets.

Validity of the findings

The proposed GPU implementation in novel. The reported results could help researchers run experiments faster in various domains.

---

## Round 0.2 · accepted · Accept

· Academic Editor

Accept

Reviewers have accepted your comments

·

Basic reporting

No comment

Experimental design

No comment

Validity of the findings

No comment

Additional comments

If the authors agree, the caption of Figure 1 may need more explanation on this question:
Which arrow (left or right) corresponds to the path where the condition of the parent node is met?

Otherwise, the manuscript may be accepted as is.

·

Basic reporting

No comment.

Experimental design

No comment.

Validity of the findings

No comment

Additional comments

The current submitted updates are sufficient for paper publication.